# Semi-Supervised Classification of Dynamical Regimes in Hamiltonian Systems Using Poincaré Sections via Contrastive Loss p-Laplacian Propagation

**Elena Avdyushina**
Donetsk State University
Donetsk, Russia
elena.v.a.2023@mail.ru

**Constantin Ruchkin**
"CR Consult"
Donetsk, Russia
construchk@gmail.com

## Abstract

In this article, we study a recent graph-based semi-supervised machine learning method—Contrastive Loss p-Laplacian Propagation (CLpLP)—for classifying solutions of dynamical systems from two-dimensional Poincaré section data. The combination of p-Laplacian label propagation and contrastive learning provides competitive classification accuracy while requiring fewer labeled samples and less training time than classical graph-based approaches.

The proposed method consists of two stages. In the first stage, contrastive learning is used to construct an informative embedding space. In the second stage, labels are propagated on a graph built in that embedding space. The preliminary contrastive training allows for better separation of classes.

This work includes: development of an automated system for analyzing dynamical regimes from Poincaré sections; graph discretization of the solution space and formulation of the classification task in a graph-based semi-supervised setting; adaptation of the two-stage CLpLP method to two-dimensional data; and regime classification for a classical mechanical system (rigid-body integrability), where regular and multi-regular orbits are identified, localized, and classified (3 regular cases and 6 multi-regular cases).

To optimize the loss function, we use Newton-based optimization together with a convolutional neural network in the contrastive-learning stage. The method achieves more than 85% accuracy even with a small number of labels per class, while accounting for accumulation errors and time constraints. This result is sufficient for our broader objective of building an automated real-time system for classifying solutions of dynamical systems.

The proposed approach is designed for scenarios with large collections of observations (trajectories/sections), costly expert labeling, and the need to construct regime maps over wide parameter ranges.

## 1 Introduction

The problem of automatically detecting integrable cases and regular regimes in dynamical systems is one of the key challenges in the qualitative analysis of nonlinear dynamics. Classical approaches to integrability detection, including the search for first integrals, resonance analysis, the construction of stability diagrams, and the study of the onset of chaos using various indicators, generally require substantial expert involvement, manual parameter tuning, and computationally expensive procedures. These difficulties become especially pronounced when investigating families of multiparameter systems, where one needs to distinguish quickly and reliably between regions of regular, quasi-regular, resonant, and chaotic behavior.

Despite the existence of a well-developed mathematical framework, primarily KAM theory, the practical solution of this problem still relies in most cases either on geometric and numerical meth-

ods or on analytical results for special cases. The use of artificial intelligence methods, and machine learning in particular, in this area remains limited, partly because of the specific nature of the problem itself. One reason is that many key properties of dynamical regimes are qualitative and geometrically interpretable, such as trajectory closure, cloud-like point structures, or asymptotic convergence. Such semantically meaningful features are difficult to represent directly as standard numerical characteristics, which complicates the use of classical machine learning methods, especially in their standard vector-based formulation.

For this reason, the development of new machine learning methods and the adaptation of existing ones in a way consistent with the mathematical and geometric aspects of KAM theory appears to be a highly promising direction. However, the number of studies in this area remains small. Some local formulations have been considered, for example, in astronomy, quantum computing, and other applied domains, whereas the problem of automatically identifying global integrable cases remains largely unexplored.

## 2 MAIN IDEA

In this work, we consider the problem of *semi-supervised classification of dynamical regimes in Hamiltonian systems* based on *Poincaré section data*. Classical approaches to detecting regular and chaotic regimes include Lyapunov indicators, Poincaré sections, intermittency analysis, sensitivity to initial conditions, attractor detection, and related methods. In most existing studies, machine learning is applied to a dynamical system represented as a time series.

In the present formulation, unlike approaches based on time series, the object of analysis is a *discrete orbit representation on a Poincaré map*, that is, a set of trajectory intersection points with a fixed surface of section. Each orbit must be assigned to a class corresponding to a particular dynamical regime, such as *regular*, *multi-regular*, *resonant*, or *chaotic*, depending on the chosen model and labeling scheme.

It is important to emphasize that, in this setting, we classify not abstract point sets as purely geometric objects, but rather *structured orbits* that reflect intrinsic properties of the underlying dynamical system. Regularity and chaos are treated not as external visual patterns, but as characteristics of motion itself. In this sense, a closed or nearly closed orbit corresponds to a *regular regime*, whereas a cloud of points filling a region of the section corresponds to a *chaotic regime*. At the same time, the visual diversity of closed trajectories remains substantial even for conservative dynamical systems. This is precisely what makes the problem difficult to formalize: the model must account not only for geometry, but also for its dynamical meaning.

The aim of the work is to construct a classifier capable of distinguishing dynamical regimes from the geometry of orbits on a Poincaré section *when only a small number of labeled examples is available*. Such a formulation is natural for Hamiltonian dynamics, where obtaining reliable labels often requires expert inspection of phase portraits, the computation of additional dynamical indicators, or long numerical integration. Therefore, we consider the problem in a *semi-supervised setting* (GSSL), where a small labeled set is combined with a much larger unlabeled one, and labels are inferred from geometric similarity in a learned representation space.

The difficulty of the problem is determined by the fact that an orbit on a Poincaré section is *not a fixed-length feature vector*, but a geometric object of variable complexity. Classification based on raw point coordinates is often insufficient. Orbits from the same class may differ substantially in scale, point density, local curvature, and degree of filling of invariant structures, while orbits from different classes may look visually similar, especially near transitions between regular and chaotic motion or in the presence of thin resonant structures. In addition, the representation of an orbit depends on the number of discretized points, the accuracy of numerical integration, and the choice of the section itself. Therefore, it is necessary to construct a data representation that preserves the geometry relevant to the type of motion while remaining robust to these variations.

The choice of a particular *conservative mechanical system* is motivated by the existence of *well-known classical integrable cases*, which can serve as *test examples* for validating the proposed method. This makes it possible to evaluate the classification results not only computationally, but also from the viewpoint of physical and mathematical interpretability. Thus, the input data consist of *orbits represented by points on Poincaré sections*, while the output consists of *labels of dynamical*

*regimes*. The specificity of the problem is determined by three factors: the geometric complexity of the input objects, the limited amount of labeled data, and the need to distinguish regimes having clear physical meaning in the framework of Hamiltonian dynamics.

Although this paper considers only *one specific classification approach*, it is part of a broader research program aimed at building an *automated system for the search for integrable cases* Ruchkin (2014). In this sense, the proposed method should be regarded as one module of a more general framework in which machine learning, geometric analysis of Poincaré sections, and semi-supervised learning are combined into a unified tool for identifying regular structures and potentially integrable regimes.

In addition, the approaches developed in this work may also prove useful in related problems of classical mechanics, fluid mechanics, astrophysics, and astronomy, where one needs to identify complex geometric and dynamical structures in large-scale datasets.

## 3 LITERATURE REVIEW OF MACHINE LEARNING FOR AUTOMATIC REGIME CLASSIFICATION FROM POINCARÉ MAPS

Automatic regime classification from Poincaré maps relies on the observation that different types of motion in Hamiltonian and near-Hamiltonian systems appear as persistent geometric and topological structures on the section: quasi-periodic invariant curves (tori), resonant island chains, separatrix structures, and stochastic (ergodic) regions. Hence, most approaches can be organized as a pipeline: construct the Poincaré map and extract orbits; choose a representation (point clouds, contours, topological primitives, graph-based descriptions); train a classification model (interpretable or deep), often accounting for invariances and constraints induced by the underlying physical structure.

Early work on visualization and analysis of area-preserving maps established a "vocabulary" of structures that naturally become objects for feature-based descriptions and (semi-)automatic labeling. In particular, it has been shown that the visual organization of Poincaré maps typically includes fixed (or periodic) points embedded in island chains, invariant manifolds, and regions of ergodic behavior, and that this topological organization largely determines the separability of regimes. These results are important for ML formulations because they specify *which* geometric/topological primitives are worth extracting prior to learning, in order to reduce the classifier's sensitivity to sampling density, noise, and fragmentation of point clouds.

The most direct ML approach for this topic is to train a classifier on orbits (sets of intersection points) and predict the regime/structure type Kamath (2024). Kamath formulates orbit classification on a Poincaré map into four classes (*quasiperiodic*, *separatrix*, *island chain*, *stochastic*) and shows that carefully engineered features aligned with the orbit point-cloud geometry are crucial. The choice of interpretable models (e.g., decision trees) is also significant, as it supports error diagnosis, identification of ambiguous regions between classes (e.g., borderline cases between quasi-periodic motion and separatrix dynamics), and iterative refinement of both features and labels. For automatic regime classification from Poincaré maps, this provides a practical template: "map structure → features → interpretable classifier → error analysis and rule/label refinement."

A parallel line of work classifies regimes without explicitly constructing a Poincaré map, using observed time series instead Celletti et al. (2022). Celletti *et al.* demonstrate the effectiveness of convolutional architectures for classifying motion types from time series "without a priori knowledge of the dynamics," where training labels are obtained using classical dynamical indicators, and the ability to transfer to "similar but not identical" systems is examined. For articles focused on Poincaré-based classification, this is relevant for two reasons: (i) such models provide an alternative (sometimes computationally cheaper) source of automatic labeling for Poincaré structures; (ii) they highlight the role of a "weak teacher" (chaos/regularity indicators) that turns computationally intensive phase-space cartography into a scalable classification problem.

In dynamical systems and chaos, interpretability is often not optional but a scientific requirement: it is important to understand which system properties the model exploits and how transferable it is. Wang and Li propose a dynamics-oriented deep-learning scheme (DSDL), motivating it as a compromise between long-horizon predictive accuracy and transparency/interpretability through connections to nonlinear dynamics theory Wang & Li (2024). In a related but more explicitly "structure-physical" formulation, Desai *et al.* introduce port-Hamiltonian neural networks for ex-

plicit non-autonomous systems by embedding a formalism that can jointly account for dissipation and time-dependent forcing Desai et al. (2021). In the context of Poincaré-based regime classification, these works justify the value of inductive biases—either via interpretable model components or via physical constraints (energy-based structure)–which can improve robustness and reproducibility under parameter changes and incomplete observations.

We also note a methodologically important idea: "orbit classification as classification of equivalence classes." Revis, Zakaryan, and Raissi consider orbits generated by local transformations (local complementation and local scaling) and represent each equivalence class as an *orbit graph*, where nodes correspond to states and edges correspond to elementary transformations Revis et al. (2025). Although the application domain is quantum graph states, the construction is transferable to Poincaré-based classification: one can explicitly define admissible transformations of an orbit representation (e.g., phase shifts on the section, coordinate rotations/scalings, point subsampling, or re-orderings that preserve invariants) and then build either invariant features or a graph model (e.g., via connectivity/distance statistics), so that the classifier distinguishes regimes rather than artifacts of parameterization and visualization. This perspective is particularly useful when building robust ML systems on Poincaré maps, where the same regime may appear differently due to section choice, integration density, and noise.

Overall, the following ML-oriented pipeline for automatic regime classification from Poincaré maps: (1) extract structurally meaningful primitives (fixed/periodic points, island chains, separatrices, ergodicity statistics) as the basis for features; (2) use interpretable orbit classification (feature-based) for scientific explainability and iterative label debugging; (3) employ complementary deep-learning on time series/indicators as a source of large-scale (possibly weak) labels and as a tool to assess transferability; (4) improve robustness through interpretable/physics-structured models and through explicit invariance design (up to orbit-graph representations). Together, these elements naturally motivate the development of a method that classifies regimes on a Poincaré map while retaining interpretability, representation robustness, and transferability across related dynamical systems.

# 4 Application of GSSL to Orbit Classification

## 4.1 General Framework for Classifying Regular and Chaotic Solutions

The existing methods of KAM theory provide a clear characterization of a dynamical system and predict its behavior under given initial conditions, whether regular, quasi-regular, or chaotic. However, a comprehensive analytical study is challenging in practice because of high dimensionality and many parameters. Consequently, the solutions obtained are often highly localized and approximate. When numerical and computational methods are employed, errors and cumulative numerical inaccuracies arise, making the investigation of global regular cases particularly difficult Neimark et al. (2005); Petalas et al. (2008); Kondratiev & Lyaptsev (2012).

A useful approach for numerical studies of phase space, which consists of non-intersecting trajectories, is through Poincaré sections. These sections reduce the system dimension by one. Dynamical systems of third and fourth order are of particular interest, as their Poincaré sections produce graphical representations either on a plane or in three-dimensional space. When phase-flow points form a curve, the system exhibits regular behavior (periodic or multi-periodic), as observed in Hamiltonian systems.

A major research focus is the reconstruction and analysis of global Poincaré sections, which provide insights into all possible motions of the system. Although Poincaré sections are approximated using numerical integration over a fixed time interval, they still offer valuable insights into the overall behavior of Hamiltonian systems. The resulting phase portraits of two- and three-dimensional Poincaré sections can be analyzed using statistical or deterministic pattern-recognition techniques.

In modern studies, the significance of new results in dynamical systems often relies on computational analysis and advanced numerical techniques. Of particular interest are machine learning methods such as label propagation for solving classification tasks in dynamical systems. This represents a promising technique for the automatic detection of regular and chaotic behavior in such systems.

## 4.2 Hamiltonian System

Let a dynamical system of Hamiltonian type and even order be given, for example, order two. Assume that the system possesses an energy integral, i.e., the Hamiltonian is defined as $H(q, p), (q, p) \in \Omega \subset \mathbb{R}^{2n}$, and the system is described by Hamilton's equations:

$$\dot{q} = \frac{\partial H(q, p)}{\partial p}, \quad \dot{p} = -\frac{\partial H(q, p)}{\partial q}. \tag{1}$$

Assume that the system admits a global or local Poincaré section $S$. Choose a smooth section $S$ in phase space defined by $g(q, p) = 0, \nabla g(q, p) \neq 0$.

For each initial condition $(q(0), p(0))$, consider the phase trajectory $\gamma(t) = (q(t), p(t))$ and record its intersections with $S$. As a result, we obtain a set of points $\mathcal{X} = \{x_i\}_{i=1}^{N} \subset S \subset \mathbb{R}^2$.

Assume that the trajectories are divided into $K$ classes corresponding to different dynamical regimes. For a subset of indices $L \subset \{1, 2, \ldots, N\}$, labels are provided: $y_i \in \{1, 2, \ldots, K\}, \quad i \in L$. It is assumed that if the points $x_i$ and $x_j$ (corresponding to close initial conditions) are nearby, then their trajectories belong to the same class.

The set of points forms a trajectory structure of the phase space, which in numerical integration is discrete and can be divided into classes using machine learning methods.

The goal of the article is to construct a classifier $f \colon S \to \{1, 2, \ldots, K\}$, which assigns to each unlabeled point $x_i$, for $i \in U = \{1, \ldots, N\} \setminus L$, a label $\hat{y}_i = f(x_i)$. To this end, we employ a graph-based semi-supervised learning method based on a graph model.

Thus, using the Hamiltonian equations, the construction of a global (or local) Poincaré section, and a graph-based model, we obtain a rigorous mathematical formulation of the semi-supervised classification problem for the trajectory structure of the phase space using the p-Laplacian label propagation method.

## 4.3 Example of a Mechanical System

The problem of the motion of a rigid body with a fixed point is considered in the classical framework, addressing the direct problems of mechanics. Given the static, kinematic, and dynamic (structural) parameters of a rigid body with a fixed point and the initial conditions of motion, it is necessary to determine its trajectory in space at any given moment, classify the type of motion, and establish the nature of the body's dynamics.

A mathematical model describing the motion of a free rigid body in a mobile reference frame is given by a system of six ordinary differential Euler equations:

$$J\dot{\omega} = J\omega \times \omega + r \times \nu, \quad \dot{\nu} = \nu \times \omega, \tag{2}$$

where $J = \mathrm{Diag}(A, B, C)$ represents the inertia tensor, $\omega = (p, q, r)$ is the angular velocity of the body in the mobile frame, $\nu = (\nu_1, \nu_2, \nu_3)$ is the unit vertical vector, and $r = (r_1, r_2, r_3)$ is the vector from the fixed point to the center of mass of the body.

Equation (2) determines the phase space $R^{12}$ as $R^6(\omega, \nu) \times R^6(J, r)$, defining the family of possible (both regular and chaotic) trajectories of the dynamical system. The initial conditions are given by $\omega_0 = \omega(0)$ and $\nu_0 = \nu(0)$.

In the regular case, the dynamical system exhibits "well-behaved" trajectories that remain stable under small perturbations of initial conditions, allowing long-term predictability. In contrast, chaotic motion is characterized by an extreme sensitivity to initial conditions, leading to exponentially unstable trajectories. The chaotic case limits predictability to a finite time interval, known as the time horizon.

The aim of the present article is to develop an interactive computational system to solve system (2) and classify its behavior. If system (2) can be integrated explicitly, it is possible to distinguish between regular and chaotic cases.

It is known that the solution of system (2) can be reduced to quadratures if four integrals are found. However, for arbitrary values of the system parameters, only three first integrals exist.

$$H(\omega,\nu) = \frac{1}{2}J\omega \cdot \omega - r \times \nu = h, \quad G(\omega,\nu) = J\omega \cdot \nu = g, \quad I(\nu,\nu) = \nu \cdot \nu = 1. \quad (3)$$

A fourth integral exists only in three well-known cases: Euler-Poinsot, Lagrange-Poisson, and Kowalewskaya, as well as in some special cases. These cases are characterized by specific parameter sets, allowing the reduction of system (2) to a system of algebraic equations. In integrable cases, the phase space trajectories are deterministic and exhibit stable motion. For non-integrable cases, the system's trajectory cannot be expressed analytically, necessitating numerical integration using the fifth-order Runge-Kutta method. The nature of the motion is determined by analyzing phase-space trajectories. For a fixed point in $R^6(J,r)$, the first integrals (3) define a compact three-dimensional invariant manifold $Q_{h,g}^3$ in $R^6(\omega,\nu)$, which governs the system's phase flow. The spatial organization of $Q_{h,g}^3$ within $R^3(\omega)$ and $R^3(\nu)$ provides insight into the evolution of physical quantities described by equation (2).

The qualitative analysis of phase space structures relies on Poincaré sections and the theory of invariant curves. The onset of chaotic behavior is identified when the Poincaré section appears as a scattered cloud of points, forming a two-dimensional region. In contrast, regular motion corresponds to phase trajectories confined to smooth curves.

A computational program was developed for the intelligent analysis of regular and chaotic dynamics in mechanical systems, demonstrating the feasibility of this approach. An example of regular and chaotic motion identified using a three-dimensional Poincaré section is shown in Table. 1. This Poincaré section was computed using the Runge–Kutta 4(5) method for system (2) and reconstructed on the Poisson sphere.

In integrable cases, the Poincaré section is a planar region showing closed curves of different types: circles, ellipses, sets of circles, and others (Table 1). These cases correspond to regular (quasi-periodic) solutions of a nonlinear dynamical system. We use machine learning to classify them. Table 2–3 show examples of curve extraction and classification for regular and multi-regular orbits.

Table 1: All cases: regular and multi-regular orbits in Poincaré plots (1, 2), and regular, chaotic, and resonance orbits (3).

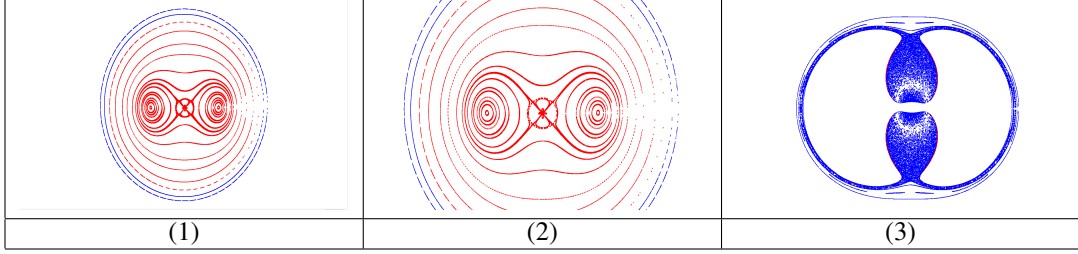

| (1) | (2) | (3) |

Table 2: Regular orbits. Cases 1, 2, and 3

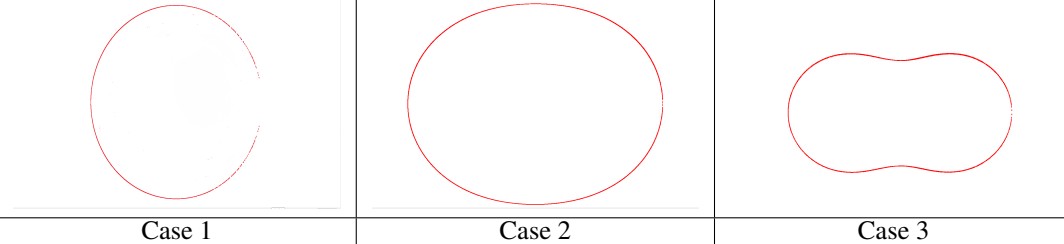

| Case 1 | Case 2 | Case 3 |

Table 3: Multi-regular orbits: Cases 1–6.

| | | |
|---|---|---|
| Case 1 | Case 2 | Case 3 |
| Case 4 | Case 5 | Case 6 |

## 4.4 TASK OF TRAJECTORY CLASSIFICATION

We consider the problem of classifying trajectory datasets obtained from two-dimensional Poincaré sections. The data are discrete and may contain several classes. We use a graph-based semi-supervised p-Laplacian propagation method for this task because only a small number of labeled points are available initially.

We therefore adopt a GSSL formulation in which the first step is to build a graph on the dataset. For the set of vertices $X = \{x_1, x_2, \ldots, x_n\}$ of an undirected graph $G$, the first $m$ vertices receive the corresponding labels $\{y_1, y_2, \ldots, y_m\}$. The number of labeled points is less than the total, i.e., $m < n$. The task of graph-based semi-supervised learning is to propagate labels from the initial vertices to all others, that is, to infer labels for $\{x_{m+1}, x_{m+2}, \ldots, x_n\}$.

General Steps of the GSSL Method. The key steps in the Graph-Based Semi-Supervised Learning (GSSL) method follow the framework outlined in Zhu (2005).

The initial step involves constructing a weighted graph by computing similarity measures between data points. Common techniques include geometric distances, Gaussian kernels, and K-Nearest Neighbor (KNN)-based distances. The choice of weighting method depends on the data distribution. In particular, KNN-based weighting often performs well when classes overlap or are closely positioned Zhu (2005). A properly weighted graph helps label propagation remain consistent with the underlying data structure, improving classification accuracy. Label initialization selects a subset of representative points from each class to serve as labeled references. To avoid bias, it is advisable to allocate an equal number of labeled samples across classes. In practice, labels can be selected by random sampling, density-based sampling, or uncertainty-based selection. Proper initialization is critical for accurate and stable propagation. A penalty function is formulated based on the Dirichlet energy over the graph. This function encourages smoothness while preserving the information from the initially labeled nodes. Additional terms can impose regularization constraints or boundary conditions, improving stability and robustness. Common regularization methods such as $\ell_2$ are used to prevent overfitting and mitigate noise. Minimizing the penalty function over the graph corresponds to solving the p-Laplacian equation. For $p = 2$, this reduces to solving the Laplace equation. The numerical solution of these equations forms the basis of label propagation. In this research, we employ Newton optimization to minimize the penalty function. This momentum-based method accelerates convergence, improves computational efficiency, and yields more stable updates than standard gradient-based methods. Following these steps, the GSSL framework enables effective label propagation and classification even when labeled data are sparse. This approach is particularly well-suited to complex data distributions, where graph-based methods provide a flexible and scalable alternative to traditional supervised learning.

## 5  Contrastive Loss p-Laplacian Propagation

The proposed contrastive semi-supervised p-Laplacian method combines the advantages of contrastive learning and semi-supervised strategies to improve classification quality Chen et al. (2020); He et al. (2020); Grill et al. (2020); Caron et al. (2020); Zbontar et al. (2021); Zhu (2005); Chapelle et al. (2009); Calder et al. (2020).

**1. Contrastive learning using both labeled and unlabeled data:** In the first stage, two augmented versions are created for each example (regardless of whether it is labeled). After passing through the encoder, the resulting representations are normalized and projected into a lower-dimensional space. For each anchor example, a set of positive examples (augmentations of the same object or examples with the same label) is formed, while all other examples constitute the negative set.

**2. Adaptive weighting based on the p-Laplacian graph model:** The contribution of each positive example to the loss function is adjusted by an additional weighting coefficient calculated using graph-based weights motivated by p-Laplacian regularization, which allows us to account for local geometry and the complexity of individual examples Calder et al. (2020); Garcia-Cardona et al. (2013).

**3. Semi-supervised mode:** If additional unlabeled information is available, it is integrated through pseudo-labeling, enabling the use of all available data Zhu (2005); Chapelle et al. (2009).

In summary, the method consists of two stages: learning representations with a modified contrastive loss function and training a classifier on the frozen representations.

Let $f$ be the encoder that maps a dataset sample to its representation in the latent space, and let $g$ be the classifier that maps the latent representation to the class probabilities. We aim to learn the optimal encoder $f^*$ and classifier $g^*$ by minimizing the following hybrid loss:

$$(f^*, g^*) = \arg\min_{f,g} \mathcal{L} = \arg\min_{f,g} \left[ \mathcal{L}_{CL}(f, \mathcal{D}^U) + \lambda \mathcal{L}_{LP}(f, \mathcal{D}^L, \mathcal{Y})] \right], \tag{4}$$

where $\mathcal{L}_{CL}(f, \mathcal{LP}^U)$ is the Contrastive loss net (CNN), $\mathcal{L}_{LP}(f, \mathcal{D}^L, \mathcal{Y})$ is the p-Laplacian Propagation loss, and $\lambda$ are the hyperparameter that balance the contributions of the two losses. The learned models $f^*$ and $g^*$ can accurately predict labels for new samples.

We will apply a two-stage approach, which involves the use of two phases: preprocessing and fine-tuning.

### 5.1  Contrastive learning in computer vision

Previous investigations include several articles.

**SimCLR:** Chen et al. (2020) Chen et al. (2020) proposed SimCLR - a simple yet effective architecture for learning visual representations using a large number of negative examples. The authors demonstrated that with the proper choice of augmentations and an increased batch size, competitive results can be achieved compared to fully supervised methods.

**MoCo:** He et al. (2020) He et al. (2020) introduced Momentum Contrast (MoCo), which employs an exponential moving average of the encoder parameters to form a dynamic queue of negative examples. This allows models to be trained even with relatively small batches. Later, improved versions, such as MoCo v2, were proposed.

**BYOL:** BYOL (Bootstrap Your Own Latent) (Grill et al., 2020) Grill et al. (2020) proposed a method that enables training representations without explicitly using negative examples. Instead, the model uses two network branches - an online branch and a target branch - with the target network updated via exponential moving average.

**SwAV:** Caron et al. (2020) Caron et al. (2020) presented a method based on on-the-fly clustering, where each image is assigned a cluster label generated on the fly. This approach combines the advantages of contrastive learning and clustering analysis, which is especially useful when the batch size is limited.

**Barlow Twins:** Barlow Twins (Zbontar et al., 2021) Zbontar et al. (2021) propose to minimize redundancy between the representations of different augmentations of the same image by using a correlation matrix that is driven toward a diagonal form, thereby reducing redundant information.

## 5.2 Algorithm for Contrastive Learning

Below is the algorithm for the Contrastive learning method. From the training dataset, we extract a batch of $n$ examples, $\{(x_i, y_i)\}_{i=1}^n$. If there are unlabeled data, pseudo-labels are generated.

**Step 1. Forming the batch** For each example $i$, two augmented versions are created (5):

$$x_i^{(1)} = \mathrm{Aug}(x_i), \quad x_i^{(2)} = \mathrm{Aug}(x_i). \tag{5}$$

**Step 2. Computing the representations** We use an encoder $\mathrm{Enc}$ and we normalize the obtained features (6):

$$r_i^{(j)} = \mathrm{Enc}\big(x_i^{(j)}\big), \;\; z_i^{(j)} = \frac{r_i^{(j)}}{\big\|r_i^{(j)}\big\|}, \quad j = 1, 2. \tag{6}$$

**Step 3. Projection** The datasets are passed through a projection network. Then we normalize again (7):

$$p_i^{(j)} = \mathrm{Proj}\big(z_i^{(j)}\big). \; \hat{p}_i^{(j)} = \frac{p_i^{(j)}}{\big\|p_i^{(j)}\big\|}, \quad j = 1, 2. \tag{7}$$

**Step 4. Forming positive and negative pairs** For each anchor example $i$, we determine the set of positives (8):

$$P(i) = \{ \, j \; \mid \; y_j = y_i \, \}, \tag{8}$$

which can include augmentations of the same example or other examples with the same (real or pseudo-) label. The negative set is defined as (9):

$$N(i) = \{ \, a : y_a \neq y_i \, \}. \tag{9}$$

**Step 5. Calculating the Contrastive Loss Function with Poisson Weighting** For each anchor $i$ and a positive example $p \in P(i)$, we compute the local loss (10):

$$L_{i,p} = -\log\left(\frac{\hat{p}_i \cdot \hat{p}_p}{\sum_{c \in \mathcal{A}(i)} \exp\big(\hat{p}_i \cdot \hat{p}_c \,/\, \tau\big)}\right) \; * \; \frac{\exp\big(-\lambda\big) \cdot \lambda^{k(i,p)}}{k(i,p)!}, \tag{10}$$

where: - $\tau$ is the temperature parameter; - $\mathcal{A}(i) = \{ \, 1, 2, \ldots, 2N \, \}$ represents the set of many other examples in the batch; - $k(i, p)$ is the number of observations of this "type" in the group, and $\lambda$ is a parameter that sets the mean value.

The average loss over all positive pairs for a given anchor $i$ can be expressed as (11):

$$L_{wup}(i) \;=\; \frac{1}{|P(i)|} \sum_{p \in P(i)} L_{i,p}. \tag{11}$$

Finally, the overall batch loss is typically the average of $L_{wup}(i)$ across all anchors $i$ in the batch.

**Step 6. Backpropagation and Parameter Updates** Using the chosen optimizer, the error is propagated backward, and the network parameters are updated based on the computed loss function $L_{\mathrm{sup}}$.

**Step 7. Classifier Training** After representation learning is completed, the final classifier is trained on frozen representations $z$, usually using a standard cross-entropy loss. In our case, we then apply p-Laplacian learning, which is discussed in the following section. We rely on ideas presented in this article.

## 5.3   Algorithms for $p$-Laplacian Learning

We now present algorithms with variational $p$-Laplacians on a graph and a Newton-like optimization method.

Since $J_p$ is smooth and convex, it is natural to use Newton's method to minimize $J_p$. We give here the explicit details of the Newton iteration for minimizing $J_p$. It is useful to first rewrite the function $E_p(u)$ using vector notation. Let $X = \{x^1, \ldots, x^n\} \subset \mathbb{R}^d$, where $O = \{x^{n+1}, \ldots, x^{n+m}\}$ is the observation set. We define $u_i = u(x^i)$ and set $u = (u_1, \ldots, u_n) \in \mathbb{R}^n$. Similarly, set $w_{ij} = w_{x^i x^j}$, $f_i = f(x^i)$, $g_j = g(x^{n+j})$, $f = (f_1, \ldots, f_n) \in \mathbb{R}^n$, and $g = (g_1, \ldots, g_m) \in \mathbb{R}^m$. Then, subject to the constraints in (12), we can write

$$J_p(u) = \frac{1}{p} \left( \sum_{i=1}^{n} \sum_{j=i+1}^{n} w_{ij} \, |u_i - u_j|^p + \sum_{i=1}^{n} \sum_{j=1}^{m} w_{i,j+n} \, |u_i - g_j|^p \right) + \sum_{i=1}^{n} f_i u_i. \tag{12}$$

Newton's method corresponds to the iteration

$$u^{k+1} = u^k - \left[ \nabla^2 J_p(u^k) \right]^{-1} \nabla J_p(u^k). \tag{13}$$

For notational convenience, define $a_{ij}(u) = w_{ij}|u_i - u_j|^{p-2}$ and $b_{ij}(u) = w_{i,j+n}|u_i - g_j|^{p-2}$, and

$$d_i(u) = \sum_{j=1}^{n} a_{ij}(u) + \sum_{j=1}^{m} b_{ij}(u). \tag{14}$$

Then, we write $A(u) = (a_{ij}(u))_{ij} \in \mathbb{R}^{n \times n}$, $B(u) = (b_{ij}(u))_{ij} \in \mathbb{R}^{n \times m}$ and $D(u) = \mathrm{diag}(d_i(u)) \in \mathbb{R}^{n \times n}$.

In this notation, we compute

$$\nabla J_p(u) = L(u)u - B(u)g + f \quad \text{and}$$

$$\nabla^2 J_p(u) = (p-1)L(u),$$

where $L(u) := D(u) - A(u)$, and thus the Newton update is given by

$$u^{k+1} = \frac{p-2}{p-1} u^k + \frac{1}{p-1} L(u^k)^{-1} \big[ B(u^k)g - f \big]. \tag{15}$$

The inversion of $L(u^k)$ is performed with an iterative method, such as the preconditioned conjugate gradient or Newton. The matrix $L(u^k)$, being a graph-Laplacian, is always positive semi-definite. If the graph is connected and $u$ is nondegenerate, then it is also non-singular.

## 5.4   Algorithm: Contrastive Loss p-Laplacian Learning

We propose combining contrastive learning and semi-supervised learning, with p-Laplacian propagation used for final classification.

It can be applied when only a small amount of labeled data is available. However, the method also has some drawbacks related to balancing initial labels and class distributions. To avoid these drawbacks, we will assume that the data and initial labels are balanced across the classes.

## 5.5 CLASSIFICATION RESULTS

We obtained classification results for regular and multi-regular cases. We tested the algorithm on 20 different images with two labeled trajectories per class. The algorithm achieved an accuracy greater than 85%. The classification results are shown in Fig. 1.

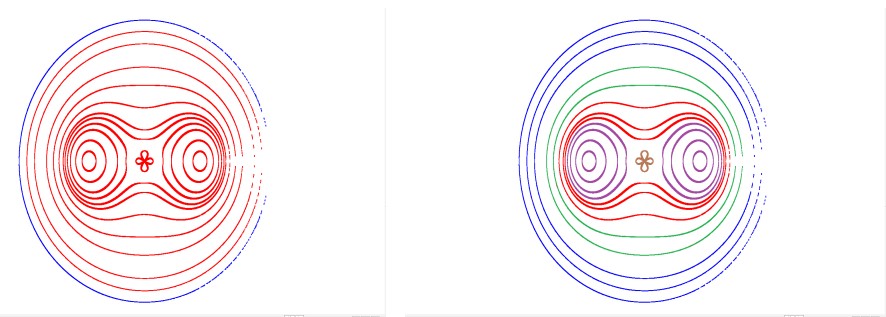

Figure 1: Regular and multi-regular orbits on Poincaré plots (1, 2): regular case 1 (blue), case 2 (green), and multi-regular cases 3 (red), 4 (violet), and 5 (brown).

---

**Algorithm 1** Contrastive Learning p-Laplacian Propagation

---

1: **procedure** CLFRAMEWORK($X, y, N, \tau, \lambda, epochs$)
2:     **for** epoch $= 1$ to $epochs$ **do**
3:         Extract batch: $\{(x_i, y_i)\}_{i=1}^N$
4:         Generate pseudo-labels for unlabeled data
5:         Compute augmentations: $x_i^{(1)} = \text{Aug}(x_i)$, $x_i^{(2)} = \text{Aug}(x_i)$
6:         Encode representations: $r_i^{(j)} = \text{Enc}(x_i^{(j)})$ and Normalize: $z_i^{(j)} = \frac{r_i^{(j)}}{\|r_i^{(j)}\|}$
7:         Project: $p_i^{(j)} = \text{Proj}(z_i^{(j)})$ and Normalize projections: $\hat{p}_i^{(j)} = \frac{p_i^{(j)}}{\|p_i^{(j)}\|}$
8:         Form positive and negative pairs: $P(i) = \{j \mid y_j = y_i\}$, $N(i) = \{a \mid y_a \neq y_i\}$
9:         Compute Loss:    $L_{i,p} = -\log \frac{\exp(\hat{p}_i \cdot \hat{p}_p / \tau)}{\sum_{c \in \mathcal{A}(i)} \exp(\hat{p}_i \cdot \hat{p}_c / \tau)} * w_{i,p} = \frac{\exp(-\lambda)\lambda^{k(i,p)}}{k(i,p)!}$
10:         Compute average loss per anchor:    $L_{wup}(i) = \frac{1}{|P(i)|} \sum_{p \in P(i)} L_{i,p}$
11:         Update parameters by backpropagation using $L_{\text{sup}} = \frac{1}{n} \sum_{i=1}^n L_{wup}(i)$
12:     **end for**
13: **end procedure**
1: **procedure** PLPFRAMEWORK($y, T$)
2:     Initialization $Y^0 = [y_1, y_2, \ldots, y_m]$
3:     Compute $W = \text{kNN}$
4:     Compute Degree Matrix D: $d_{ii} = \sum_{j=1}^n w_{ij}$
5:     Compute Laplacian: $\boldsymbol{L} = D - W$
6:     Compute Average Label Vector: $\bar{y} = \frac{1}{m} \sum_{j=1}^m y_j$
7:     Construct Matrix: $F^m = [U - \bar{y}, \text{Zero}(2, n - m)]^T$
8:     Initialize: $U^{\{0\}} = \text{Z}(n, 2)$
9:     **for** $k = 1$ to $T$ **do**
10:         Update formulas (14) and (15)
11:     **end for**
12:     Label Assignment: $l_{max}^k = \arg\max_{1 \leq j \leq 2} U_j^k$
13:     **return** $u_l = [l_1, l_2, \ldots, l_n]$
14: **end procedure**

---

## 6 CONCLUSION

In this paper we consider the problem of classifying periodic and multi-periodic orbits of dynamical systems using a graph-based semi-supervised p-Laplacian propagation method. The detection of periodic orbits is carried out by analyzing Poincaré sections of phase space on the plane (2D datasets). We propose a modification of graph-based semi-supervised learning via the p-Laplacian equation. To minimize the loss function, we employ the Newton algorithm, which demonstrates better results than gradient descent. With this approach and modification, the task of classifying periodic solutions of the dynamical system is solved with sufficient accuracy even in real time, taking into account accumulation errors and time constraints.

p-Laplacian propagation offers advantages over traditional Laplace-based approaches, requiring significantly fewer labeled samples to achieve the desired classification accuracy while also reducing computational time.

This work includes: development of an automated system for analyzing dynamical regimes from Poincaré sections; graph discretization of the solution space and formulation of the classification task in a graph-based semi-supervised setting; adaptation of the two-stage CLpLP method to two-dimensional data; and regime classification for a classical mechanical system (rigid-body integrability), where regular and multi-regular orbits are identified, localized, and classified (3 regular cases and 6 multi-regular cases).

To optimize the loss function, we use Newton-based optimization together with a convolutional neural network in the contrastive-learning stage. The method achieves more than 85% accuracy even with a small number of labels per class, while accounting for accumulation errors and time constraints.

The proposed approach for investigating dynamical systems expands the possibilities of analytical and numerical use of KAM theory and is the next step toward building a computer system for fully automated investigation of dynamical systems.

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
