# OpenReview forum: "Semi-Supervised Classification  of Dynamical Regimes in Hamiltonian Systems Using Poincaré Sections via Contrastive Loss  p-Laplacian Propagation"
_mathai.club/MathAI/2026/Conference — 2026 Oral_

### Official Review · Reviewer_PJ32 · 2026-03-11
**Assessment of a Graph-Based Approach for Poincaré Section Analysis**

**Rating:** 6
**Confidence:** 4

**Review:**

The paper addresses the problem of identifying the type of solutions of dynamical systems through the analysis of Poincaré sections. The proposed approach consists of projecting trajectories onto the Poincaré surface, constructing a graph from the obtained points, and applying semi-supervised learning methods to identify curve structures corresponding to different types of orbits.

The literature review is presented correctly and covers the main existing approaches to solving this problem. However, only a limited number of recent publications are discussed, which may indicate that this research direction has not gained wide popularity in the recent literature.

The mathematical formulation of the problem is presented correctly, and the chosen model problem is illustrative and suitable for demonstrating the applicability of the proposed method. The authors emphasize a scenario with a small number of labeled samples, interpretability of the approach, and claim computational efficiency of the proposed method.

It is worth noting the use of representation learning techniques, which represent an important and actively developing area in modern machine learning. The proposed data augmentation scheme for preparing training samples appears to be a reasonable and useful solution.

At the same time, the choice of a graph-based model does not appear to be fully justified. In particular, the paper does not provide a comparison with alternative approaches, such as convolutional neural networks, point-cloud processing methods, or modern object detection/segmentation techniques applied to images.

The dataset used in the experiments is relatively small; however, for a proof-of-concept study this can be considered acceptable. Nevertheless, the paper does not provide an analysis of the computational complexity of the proposed algorithm (for example, in Big-O notation) either for the training stage or for the detection procedure.

The presented results also raise some questions. The regular curves are highlighted with different colors and separated into several interpretable clusters; however, results for the chaotic regime — which represents an important aspect of dynamical systems analysis — are not provided.

Finally, the paper does not discuss potential applications of the proposed method to real-world problems. In addition, the use of the term “intelligent system” in the title may be somewhat overstated, since the presented approach appears to describe a specific algorithm rather than a complete intelligent system.

---

### Official Review · Reviewer_Mjx1 · 2026-03-12
**The work focuses on the development of a graph-based semi-guided machine learning method for classifying dynamic system solutions based on two-dimensional Poincare section data. The structure of the work allows us to make a complete picture of the method and its stages, but the overview of existing methods does not look complete enough.**

**Rating:** 6
**Confidence:** 4

**Review:**

The background section covers the basics well, though it could include more recent studies. The approach is easy to interpret, and the data augmentation strategy is a smart solution. Using representation learning is also a strong point, as this is a key area in modern machine learning.
To make the paper more convincing, a few things need to be addressed:
- The authors use a graph-based model, but they do not explain why this is better than other options. A comparison with standard methods is needed.
- The dataset is small, which is acceptable for a initial test. However, the paper must analyze the computational cost for both training and detection.
- The results show clear clusters for regular curves, but there is no data for the chaotic regime. Since chaos is a major part of dynamical systems, these results are missing.
- The paper does not mention how this method could be used in real-world situations.

The core idea is valuable and the technical foundation is solid.

---

### Official Review · Reviewer_wwJG · 2026-03-12
**Review for “An intelligent system for automatic detection of integrable cases in dynamical systems using machine learning methods”**

**Rating:** 6
**Confidence:** 4

**Review:**

The paper proposes an intelligent system for automatic detection of integrable cases in dynamical systems using machine learning techniques. The authors study a graph-based semi-supervised learning method called Contrastive Loss $p$-Laplacian Propagation (CLpLP) for the classification of solutions of dynamical systems based on two-dimensional Poincaré section data.

The proposed framework consists of two stages. In the first stage, contrastive learning is used to construct an embedding space that separates different types of trajectories. In the second stage, labels are propagated across a graph constructed in the learned embedding space using a $p$-Laplacian propagation method. This combination is intended to reduce the number of labeled samples required for accurate classification.

The system is applied to a classical mechanical example involving rigid-body dynamics, where different types of orbits (regular and multi-regular) are identified and classified. According to the authors, the method achieves more than 85\% classification accuracy even when only a small number of labeled examples are available. The broader goal of the work is the development of an automated system capable of classifying dynamical regimes in near real time.

The paper addresses an interesting interdisciplinary problem at the interface between dynamical systems and machine learning. Automatic identification of dynamical regimes from Poincaré sections is a potentially useful task in computational physics and applied mathematics.

The use of semi-supervised learning is reasonable in this context, since labeled data for dynamical regimes can be expensive to obtain. The two-stage approach, combining contrastive representation learning with graph-based label propagation, is conceptually sound and aligns with modern machine learning practices.

The application to a concrete physical system provides a useful demonstration of the proposed method and illustrates how machine learning techniques can be used to assist in the analysis of dynamical systems.

While the application domain is interesting, the methodological novelty of the paper appears limited. The proposed pipeline mainly combines existing techniques, namely contrastive learning and graph-based $p$-Laplacian label propagation. The paper does not clearly demonstrate whether the specific combination of these methods leads to fundamentally new algorithmic insights or theoretical advances.

The experimental evaluation is relatively limited and focuses on a single mechanical system. It would strengthen the paper if the approach were tested on multiple dynamical systems or benchmark datasets in order to demonstrate the general applicability of the method.

In addition, some aspects of the machine learning methodology are described only briefly. For example, details about the neural network architecture used in the contrastive learning stage, the construction of the graph, and the selection of hyperparameters could be explained more clearly to improve reproducibility.

Finally, the paper would benefit from a clearer comparison with existing approaches for analyzing Poincaré sections and detecting dynamical regimes, including both classical methods from dynamical systems theory and recent machine learning approaches.

The paper presents an interesting application of machine learning techniques to the analysis of dynamical systems. The proposed system combines contrastive representation learning with graph-based label propagation in order to classify dynamical regimes from Poincaré section data. While the application is potentially useful, the methodological contribution appears to be largely an integration of existing techniques rather than the development of a fundamentally new method.

Overall, the work may be of interest to researchers working at the intersection of machine learning and dynamical systems, but the paper would benefit from a stronger demonstration of novelty and broader experimental validation.

---

### Official Review · Reviewer_2J7T · 2026-03-13
**«An Intelligent System for Automatic Detection of Integrable Cases in Dynamical Systems Using Machine Learning Methods»**

**Rating:** 6
**Confidence:** 3

**Review:**

This paper presents a hybrid CLpLP method for classifying motion types (regular/chaotic) in dynamical systems based on Poincaré sections, combining contrastive learning for feature construction and the p-Laplacian for semi-supervised label propagation on the graph. The idea is promising, and the claimed accuracy of >85% with a small amount of labeled data appears encouraging. However, the main drawback of the paper is the extremely weak experimental section: there is no comparison with baseline methods, no detailed metrics (confusion matrix), no description of augmentations and validation procedures, and a number of key algorithmic details (e.g., Poisson weighting) are not explained. For publication at a MathAI-level conference, significant refinement is required, including more robust experiments and a more detailed explanation of the method.

---

### Decision · Program_Chairs · 2026-03-14

**Decision:**

Accept (Oral)

**Comment:**

Dear Author(s),

On behalf of the Program Committee of the International Conference on Mathematics of Artificial Intelligence (MathAI 2026), we are pleased to inform you that your paper has been accepted for an oral presentation at MathAI 2026.

Your paper was evaluated through a rigorous two-stage review process involving both automated screening and expert review by members of the Program Committee. The reviewers recognized the quality and contribution of your work.

Presentation details:

- Format: Oral presentation (15–20 minutes + 5 minutes Q&A)
- Mode: You may present either in person (offline) at the conference venue in Sirius, Russia, or remotely via Zoom. Please indicate your preferred mode when confirming your participation.
- Conference dates: Marh 30 - April 3, 2026
- Website: https://mathai.club

Next steps:

1. Please confirm your participation and presentation mode by replying to this email mathai.club@yandex.ru no later than March 15, 2026 18:00 Moscow time.
2. If you plan to attend in person, the organizing committee will provide accommodation details separately.
3. Please prepare your final camera-ready manuscript according to the formatting guidelines available at https://mathai.club and upload it to OpenReview by March 15, 2026 18:00 Moscow time.

Should you have any questions regarding the program, logistics, or your presentation slot, please do not hesitate to contact us.

We look forward to your contribution to MathAI 2026.

With kind regards,

MathAI 2026 Program Committee
International Conference on Mathematics of Artificial Intelligence
https://mathai.club
OpenReview: https://openreview.net/group?id=mathai.club/MathAI/2026/Conference
Telegram: https://t.me/MathAI_club
Email: mathai.club@yandex.ru